# Interactions between the Astrocytic Volume-Regulated Anion Channel and Aquaporin 4 in Hyposmotic Regulation of Vasopressin Neuronal Activity in the Supraoptic Nucleus

**DOI:** 10.3390/cells12131723

**Published:** 2023-06-26

**Authors:** Yang Liu, Xiao-Ran Wang, Yun-Hao Jiang, Tong Li, Shuo Ling, Hong-Yang Wang, Jia-Wei Yu, Shu-Wei Jia, Xiao-Yu Liu, Chun-Mei Hou, Vladimir Parpura, Yu-Feng Wang

**Affiliations:** 1Department of Physiology, School of Basic Medical Sciences, Harbin Medical University, Harbin 150081, China2015173034@hrbmu.edu.cn (H.-Y.W.);; 2Neuroscience Laboratory for Translational Medicine, School of Mental Health, Qiqihar Medical University, Qiqihar 161006, China; 3International Translational Neuroscience Research Institute, Zhejiang Chinese Medical University, Hangzhou 310053, China

**Keywords:** aquaporin 4, glial fibrillary acidic protein, LRRC8A, volume-regulated anion channels, regulatory volume decrease

## Abstract

We assessed interactions between the astrocytic volume-regulated anion channel (VRAC) and aquaporin 4 (AQP4) in the supraoptic nucleus (SON). Acute SON slices and cultures of hypothalamic astrocytes prepared from rats received hyposmotic challenge (HOC) with/without VRAC or AQP4 blockers. In acute slices, HOC caused an early decrease with a late rebound in the neuronal firing rate of vasopressin neurons, which required activity of astrocytic AQP4 and VRAC. HOC also caused a persistent decrease in the excitatory postsynaptic current frequency, supported by VRAC and AQP4 activity in early HOC; late HOC required only VRAC activity. These events were associated with the dynamics of glial fibrillary acidic protein (GFAP) filaments, the late retraction of which was mediated by VRAC activity; this activity also mediated an HOC-evoked early increase in AQP4 expression and late subside in GFAP-AQP4 colocalization. AQP4 activity supported an early HOC-evoked increase in VRAC levels and its colocalization with GFAP. In cultured astrocytes, late HOC augmented VRAC currents, the activation of which depended on AQP4 pre-HOC/HOC activity. HOC caused an early increase in VRAC expression followed by a late rebound, requiring AQP4 and VRAC, or only AQP4 activity, respectively. Astrocytic swelling in early HOC depended on AQP4 activity, and so did the early extension of GFAP filaments. VRAC and AQP4 activity supported late regulatory volume decrease, the retraction of GFAP filaments, and subside in GFAP-VRAC colocalization. Taken together, astrocytic morphological plasticity relies on the coordinated activities of VRAC and AQP4, which are mutually regulated in the astrocytic mediation of HOC-evoked modulation of vasopressin neuronal activity.

## 1. Introduction

Neuronal activity is closely modulated by astrocytic morphological plasticity, which largely determines physical barriers and neurochemical composition around neurons [1,2]. A dramatic feature of astrocytic morphological plasticity is regulatory volume decrease (RVD), which occurs when a cell absorbs water and has intracellular hypotonicity [3]. Astrocytic RVD is regulated by many factors such as the extracellular matrix, glial fibrillary acidic protein (GFAP), aquaporin 4 (AQP4), and various ion channels [4]. GFAP filaments can scaffold the astrocytic shape; they are mainly distributed in the somata and large proximal processes of astrocytes. GFAP can serve as a platform for inter-molecular interactions and likely guides the spatial distribution of plasma membrane proteins like AQP4 [5]. In response to hyposmotic challenge (HOC), a condition similar to hyponatremia, there is an initial astrocyte swelling, followed by RVD. In parallel, GFAP filaments, and the astrocytic proximal processes that this protein scaffolds, expand transiently and then exhibit retraction. This dual astrocytic morphological plasticity is accompanied with an initial inhibition, followed by a rebound, of the electrical activity of nearby vasopressin (VP) neurons in the supraoptic nucleus (SON) [6]. Both the GFAP retraction and rebound of VP neuronal excitability also occur in response to hyperosmotic stress, which can be inhibited by blocking AQP4 [7], an astrocyte-specific water channel protein, which is essential for the volemic regulation of astrocytes [8,9].

The osmotic gradient across the plasma membrane relies on the activity of the volume-regulated anion channel (VRAC) [10], AQP4 [7], K^+^ channel Kir4.1 [11], transient receptor potential channel of the vanilloid subfamily 4 (TRPV4) [12], and several other players [13]. VRAC is an outwardly rectifying channel mainly composed of leucine-rich repeat-containing protein 8 (LRRC8). LRRC8A is an essential subunit of VRAC since its knockdown or blockade can dramatically reduce VRAC currents, RVD, and the flux of anions such as glutamate and taurine, i.e., -vesicular release [14,15,16]. AQP4-deficient astrocytes display slower cell swelling and lower VRAC activity [17], suggesting that AQP4-dependent VRAC activity is important for cell swelling. Moreover, a previous experiment revealed that increasing AQP4 expression in astrocytes did not change the expression of LRRC8A [18], suggesting that AQP4 may modulate VRAC activity but not protein expression.

It remains unknown whether VRAC interacts with AQP4 at different stages of HOC, and whether GFAP has a guiding role in VRAC distribution in the SON. It is also unknown what the functional consequence of such interactions for neuronal activity in the SON might be. These are the very issues that we address in the present work.

## 2. Materials and Methods

### 2.1. Animals

This study used young adult male Sprague Dawley rats (80–120 g) and rat pups (0- to 1-day-old) of both sexes, provided by the animal center of the Second affiliated hospital of Harbin Medical University. The use of adult males but not females can avoid the complexity introduced by the estrous cycle and oscillation of sex steroid hormones in adult females. Adult male rats were housed in groups of four in polycarbonate cages, with free access to food and water. Room temperature was maintained at 21–23 °C, with a 12 h light–12 h dark cycle. Individual dams were housed with their newborn pups. All animal procedures were in accordance with the Guideline of National Institutes of Health and approved by the Institutional Animal Care and Use Committees of Harbin Medical University (protocol no. 202-4-6, entitled “The guiding role of GFAP in astrocytic aquaporin-4 translocation and the underlying mechanism”).

### 2.2. Preparation of Acute SON Slices from Young Adult Rats

Acute hypothalamic slices were prepared using the method published previously [7]. Briefly, rats were decapitated, and hypothalamic blocks were dissected in an ice-cold slicing solution. The slicing solution contained 1/3 10% (*w*/*v*) sucrose and 2/3 regular artificial cerebrospinal fluid (aCSF) that was composed of the following (in mmol/L): 126 NaCl, 3 KCl, 1.3 MgSO_4_, 2.4 CaCl_2_, 1.3 NaH_2_PO_4_, 26 NaHCO_3_, and 10 Glucose; 305 mOsm/kg, pH 7.4, oxygenated by bubbling with a carbogen gas mixture of 95% O_2_ and 5% CO_2_. Coronal hypothalamic slices (300 μm thick) containing the SON were cut using a vibrating microtome (Leica VT1200). After pre-incubation in the regular aCSF at 35 °C for 30 min and then incubation at room temperature (22–24 °C) for 1–8 h, slices were used for patch-clamp recordings or immunohistochemistry.

### 2.3. Preparation of Astrocyte Cell Culture from Rat Pups

The primary cultures of astrocytes were prepared from the hypothalamus of newborn pups as previously described [11]. After dissection in ice-cold 0.1 mol/L phosphate-buffered saline (PBS), hypothalamic tissues, including both supraoptic and paraventricular nuclei, were digested in 0.25% trypsin at 37 °C for 20 min. Cells were dissociated by triturating tissue through a glass pipette 8–10 times in DMEM/F12 medium (SH30023.01, Hyclone, Utah) supplemented with 10% fetal bovine serum (HLJ-TF041, MRC, Changzhou), 100 units/mL of penicillin, 100 µg/mL of streptomycin, and epidermal growth factor (10 ng/mL). The dissociated cells were plated into culture flasks for 12–23 days to allow for the maturity of astrocytes, i.e., for the full expression of GFAP and AQP4 in these glial cells [11]. After purification through the shaking method [19], the cultures contained >99% of astrocytes, as confirmed by GFAP immunostaining.

### 2.4. Patch-Clamp Recordings

The patch-clamp recordings of vasopressin (VP) neuronal activity in acute SON slices followed the same procedures as described previously [7]. In brief, acute SON slices were transferred into a recording chamber and superfused with the regular aCSF or hypotonic aCSF (275 mOsm/kg, pH 7.4) at 35 °C at a flow rate of 1.5–2 mL/min. The hypotonic aCSF was prepared by reducing NaCl to 111 mmol/L from 126 mmol/L in the regular aCSF. Whole-cell patch-clamp recordings were obtained from magnocellular neuronal somata of the SON with a Multiclamp 700B amplifier (Molecular Devices, San Jose, CA, USA). The pipette solution contained the following (in mmol/L): 145 K-gluconate, 10 KCl, 1 MgCl_2_, 10 HEPES, 1 EGTA, 0.01 CaCl_2_, 2 Mg-ATP and 0.5 Na_2_-GTP, and 0.05% (*w*/*v*) biocytin (B4261, Sigma-Aldrich, Inc., St. Louis, MO, USA); pH 7.2 adjusted with KOH and osmolality of 300 mOsm/kg. Electrical signals were filtered, sampled at 5 kHz, and analyzed offline using Clampfit 10 software (Molecular Devices, San Jose, CA, USA). The firing rate was calculated every minute. A change (an increase or a decrease) in the firing rate was claimed if the frequency (Hz) of action potential (AP) discharges varied more than 3 times the standard deviation of the mean firing rate in the preceding 2 min section. Spontaneous excitatory postsynaptic currents (sEPSCs) were recorded at a holding potential of −70 mV in the whole-cell voltage-clamp configuration. The identification of VP neurons was mainly based on their phasic firing patterns in the resting condition [20] or after depolarization by 5–10 mV. As biocytin was added to the pipette solution, the SON magnocellular neurons that did not show typical phasic firing were post hoc identified as VP (vs. oxytocin) neurons using immunostaining (see below).

The methods for taking recordings from cultured astrocytes were similar to those published previously [14]. Purified astrocytes in flasks were dissociated and plated onto poly-D-lysine-coated coverslips for 24–30 h before recordings. Only one astrocyte was recorded per coverslip. Astrocytes were patched in the whole-cell configuration in isotonic bath solution containing the following (in mmol/L): 90 NaCl, 2 KCl, 1 MgCl_2_, 1 CaCl_2_, 10 HEPES, and 110 mannitol (300 mOsm/kg; pH 7.4 with NaOH). The hypotonic solution had the same ionic composition but contained 22 mmol/L instead of 110 mmol/L of mannitol, which yielded an osmolality of 200–210 mOsm/kg. Recording electrodes with a tip resistance of 3–6 MΩ were filled with a pipette solution containing (in mmol/L) 133 CsCl, 10 HEPES, 4 Mg-ATP, 0.5 Na-GTP, 2 CaCl_2_, 1 MgCl_2_, and 5 EGTA (pH 7.3 adjusted with CsOH; 290–300 mOsm/kg). The membrane potential of single astrocytes was held at −60 mV with voltage steps (500 ms duration in 3 s intervals) applied in 20 mV increments in the range of −100 to +100 mV. The relative current was calculated by dividing current values by the current value at −100 mV of the control (CTR) or 0 min HOC group.

### 2.5. Immunostaining

To examine the relative expression levels of LRRC8A and AQP4, as well as their individual spatial colocalization with GFAP in hypothalamic slices, we performed immunohistochemistry and confocal microscopy in the SON using the methods described previously [7]. Briefly, 300 µm thick brain slices were cut from hypothalamic blocks after fixation for 48 h in 4% paraformaldehyde (pH 7.4 in PBS) and triple rinsing in PBS. These slices were then dehydrated on a rotator in PBS containing increasing amounts of sucrose (10%, 20%, and 30% sucrose), associated with particular incubation times (for 1 h, 1 h, and 16h, respectively). The slices were individually placed in a plastic embedding box (7 × 7 × 1.5 mm), and their surfaces were covered with isopentane (~200 μL) before Optimal Cutting Temperature Compound (OCT, Tissue-Tek) was added. The embedding box was then quickly dipped in a container of liquid nitrogen for ~1 min and then stored at −80 °C for 3 h to a month. The slices were cut into 7-μm-thick sections on a cryostat microtome (Leica, CM 1950). The sections were mounted on adhesive-pre-coated slides, fixed in 95% ethanol for 5 min, and then, after blocking and permeation, covered with a drop of antibody-containing PBS. The primary antibodies included chicken against GFAP (1:300 dilution, Abcam, Cambridge, UK; Cat. No. ab134436), rabbit against AQP4 (1:500 dilution, Cell Signaling Technology, Danvers, MA, USA; Cat. No.59678s), mouse against LRRC8A (1:250 dilution, Santa Cruz Biotechnology Inc., Dallas, TX, USA; Cat. No. SC-517113), and mouse against VP-neurophysin (VP-NP, 1:500 dilution, Santa Cruz Biotechnology Inc., Dallas, TX, USA; Cat. No. SC-27093).

In the post hoc identification of VP neurons, slices were fixed in 4% paraformaldehyde for 4–12 h immediately after patch-clamp recordings. Antibody against VP-NP (1: 500 dilution, Santa Cruz Biotechnology Inc., Dallas, TX, USA; Cat. No. SC-390723) was used in the same procedure for immunostaining described above. Biocytin was identified through the incubation of slices with Alexa Fluor^®^ 488-conjugated Streptavidin (ThermoFisher, Cat. No. E13345).

In cultured astrocytes, LRRC8A and GFAP were examined using immunocytochemistry and confocal microscopy. Cultured astrocytes were fixed in 4% paraformaldehyde (pH 7.4 in PBS) for 30 min and permeated with 0.3% (*w*/*v*) Triton X-100 in PBS (PBST). Non-specific antibody binding sites were blocked with 5% (*w*/*v*) gelatin in PBS. Each coverslip containing astrocytes was covered with a drop of solution that contained primary antibody, and then incubated overnight at 4 °C (Santa Cruz Biotechnology Inc, No. SC-517113). After washing, astrocytes were incubated for 1.5 h at room temperature with species-matched secondary antibodies conjugated with Alexa Fluor^TM^ 488 or 555 (Invitrogen, Waltham, MA, USA; both at 1: 1000 dilution in PBST). Hoechst (bisbenzimide H 33342 trihydrochloride, Cat. No. B2261; 0.5 µg/mL for 5 min, Sigma-Aldrich, St. Louis, MO, USA) was used to label cell nuclei.

The fluorescence of different labels in slices or cell cultures was visualized with a laser scanning confocal microscope (Eclipse Ti, Nikon, Tokyo, Japan). To ensure the comparability of fluorescence measurements between different groups, slices or cell cultures from different treatments were matched with regard to the size/location of imaging fields, and the same scanning/imaging conditions were applied. No primary antibody staining was used as a control to exclude nonspecific staining.

### 2.6. Volume Measurement

To record astrocyte volemic changes, cultured astrocytes were loaded with sulforhodamine 101 (SR101, 10 μmol/L; S7635, Sigma-Aldrich, St. Louis, MO, USA), an astrocyte-specific dye [21], for 20 min at 37 °C and imaged. We obtained rapid z-stack (at 1.95 µm intervals between the planes) scans using a confocal microscope. To reduce SR101 photobleaching, the image size was set at 512 × 512 pixels and the acquisition time was about 5 s per stack.

Analysis of astrocytic volume was performed using the FIJI distribution of ImageJ [22]. In brief, stacks were concatenated into an x–y–z–t hyperstack and filtered to remove noise (median filter, 2-pixel diameters). A max-intensity z-projection was made through this hyperstack and a 2D time series was produced, in which each frame contained the full x–y extent of the soma at any given time point. An elliptical region of interest was drawn to encompass the soma narrowly across all time points in the series. The area above the threshold within this region of interest was used as a proxy for the soma volume. Volume increase at a given time point is reported as percent change from the baseline soma volume (∆V/V_0_), which is proportional to the decrease in relative fluorescence intensity (∆F/F_0_). Of note, swelling leads to dye dilution (and a decrease in fluorescence intensity), which represents the cell volume increase.

### 2.7. GFAP Fillament and Colocalization Analysis

Analyses of data followed the same methods as previously described [7,11]. To assay for GFAP expression in single optical sections, the whole frame of the image was compared using the same background level of fluorescence intensity. The length of GFAP filaments in cultured astrocytes was computed using FIJI software (version Fiji153-win-java8). In the evaluation of GFAP filaments surrounding VP neurons, only those filaments that had a length of longer than 1/2 of the VP neuronal somata were counted. The colocalization of two molecules using confocal microscopy was performed using the colorc2 plug-in of ImageJ. First, a two-color channel image was collected. Then, the channels were split into separate images (Image-Color-Split Channels). A region of interest was drawn to select the SON in hypothalamic slices or a single cell in cultured astrocytes. Finally, the colorc2 plug-in was run to obtain the Manders’ colocalization coefficient (MCC), which calculates the ratio of colocalization between two signals [23].

### 2.8. Statistical Analysis

Statistical analyses were performed using Graphpad 9.0 software. The sample size required for an individual set of experiments was pre-assessed using power analysis (set at 80% and α = 0.05). One-way repeated measures analysis of variance (ANOVA) followed by the post hoc Bonferroni, Sidak, or Dunn test for multiple comparisons was used for statistical analyses. *p* < 0.05 was considered significant. All data are presented as mean ± SEM.

## 3. Results

We provide a summary of this study’s findings in Table 1, which should be referred all the times in reading this paper.

### 3.1. Effect of Blocking Astrocytic AQP4 Activity with TGN-020 on HOC-Modulated VP Neuronal Activity in Acute SON Slices

In this study, we first confirmed the role of AQP4 in the HOC modulation of VP neuronal activity in the SON [7]. In whole-cell patch-clamp recordings (Figure 1A), HOC dually changed VP neuronal activity, i.e., it initially decreased the firing rate from 4.492 ± 0.624 Hz at 0 min HOC to 3.177 ± 0.620 Hz at 5 min HOC (n = 11, *p* < 0.05), and then the inhibitory effect subsided, i.e., there was a rebound at 10 min HOC (4.442 ± 0.799 Hz, *p* < 0.05; Figure 1A(c)). In the presence of the AQP4 blocker TGN-020 (10 μmol/L), however, HOC steadily decreased VP neuronal activity from 5.795 ± 1.573 Hz at 0 min HOC to 4.854 ± 1.532 Hz at 5 min HOC and to 3.593 ± 1.795 Hz at 10 min HOC (n = 5, *p* < 0.05; Figure 1B). These findings are consistent with previous reports [6,7].

### 3.2. Effect of Blocking Astrocytic VRAC Activity on HOC-Modulated VP Neuronal Activity in Acute SON Slices

While the findings presented above confirmed the essential role of astrocytic AQP4 in regulating VP neuronal activity in the SON [7], the contribution of astrocytic VRAC to HOC-evoked dual VP neuronal activity remains unknown. Thus, we tested the effects of phloretin, a blocker of VRAC-associated Cl^−^ currents [24], on the electrical activity of VP neurons in acute SON slices. In the presence of phloretin (30 μmol/L; added 10 min before HOC and kept throughout 10 min HOC; total of 20 min), the VP neuronal firing rate steadily decreased during HOC in 10 of the 11 recorded cells with a lack of rebound (Figure 2A). One neuron showed a progressive decrease in frequency in early, 5 min, and late, 20 min, HOC, but had an increase/”hump” in frequency at intermediate, 10 min and 15 min, time points (Figure 2A(c), the very top trace). Nonetheless, the average firing rate reduced from 5.932 ± 0.635 Hz before HOC to 5.062 ± 0.610 Hz at 5 min HOC (*p* < 0.01), and to 4.588 ± 0.680 Hz (*p* < 0.005), 4.167 ± 0.554 Hz (*p* < 0.05) and 3.648 ± 0.467 Hz (*p* < 0.01) at 10, 15, and 20 min HOC (n = 11), respectively (Figure 2A(c)).

Although phloretin strongly inhibits VRAC currents [24], it may also have other effects, such as the inhibition of glucose transporters and inflammation [25], albeit the latter on a slower time scale than our experiments. Consequently, we tested the effects of DCPIB, another VRAC inhibitor [24,26]. As shown in Figure 2B, in the presence of 20 μmol/L of DCPIB, the VP neuronal firing rate steadily decreased in all five recorded cells during HOC, without rebound. The average firing rate reduced from 5.450 ± 1.156 Hz before HOC to 3.663 ± 1.156 Hz at 5 min, and then to 1.140 ± 0.931 Hz (*p* < 0.01), 0.993 ± 0.860 Hz (*p* < 0. 01) and 1.117 ± 0.876 Hz (*p* < 0.01) at 10, 15, and 20 min (n = 5), respectively (Figure 2Bc). Of note, DCPIB also has off-target effects. However, these are different than those of phloretin; e.g., in astrocytes, it can block glutamate release via the connexin 43 hemichannel [27], inhibit the plasmalemmal glutamate transporter GLT-1 [27], activate the potassium channels TREK 1 and 2 [28], and suppress mitochondrial respiration [29]. The fact that two different VRAC inhibitors, with different off-targets, have the common inhibitory effect of HOC-evoked rebound in the firing rate indicates that the activity of VRAC is essential for the rebound of the VP neuronal firing rate. Thus, in subsequent experiments, we could use either VRAC blocker, and we selected phloretin.

### 3.3. Effect of HOC on Spontaneous Excitatory Postsynaptic Current (sEPSC) Frequency of VP Neurons in Acute SON Slices

Alongside the direct action of HOC-evoked astrocytic plasticity on the VP neuronal firing rate [7,11], astrocytes may mediate VP neuronal activity during HOC by modulating synaptic transmission, perhaps via AQP4 or VRAC activity, a phenomenon that has not been described yet. Indeed, this is the very issue that we addressed next. We recorded spontaneous sEPSCs from VP neurons during HOC in the absence and presence of TGN-020 or phloretin. Under control conditions, i.e., recording from cells bathed in regular aCSF, the sEPSC frequency of VP neurons was stable over a period of 40 min (Figure 3A). Under the condition of HOC (Figure 3B), however, the sEPSC frequency of VP neurons steadily decreased from 2.665 ± 0.419 Hz at 0 min HOC to 0.296 ± 0.144 Hz at 20 min HOC (n = 8, *p* < 0.01). Pretreatment with TGN-020 only delayed the HOC-evoked decrease in sEPSC frequency from the early (5 min) to the next intermediate (10 min) stage of HOC (Figure 3C; n = 6). However, phloretin treatment blocked the HOC-evoked decrease in sEPSC frequency (Figure 3D; n = 6). These new findings indicate that HOC can reduce sEPSC frequency and that AQP4 plays a supporting role in the early stage of the HOC-evoked decrease in sEPSC frequency, while VRAC does so in both the early and late stages.

### 3.4. Effects of Phloretin on HOC-Evoked Changes in Distribution of GFAP Filaments, AQP4 Expresion, and Their Colocalizaion in Acute SON Slices

While the results above (Figure 1, Figure 2 and Figure 3) indicate that astrocytic AQP4 and VRAC, as volume-regulating molecules, modulate VP neuronal activity, it remains unknown whether or not they achieve this feat via interactions with GFAP, a cytoskeletal element known to be a nodal mediator of astrocytic plasticity in HOC [6], which we address in this section and also in Section 3.5 and Section 3.8.

We first tested whether blocking VRAC activity with 30 μmol/L of phloretin affects the distribution of GFAP filaments, AQP4 expression, and their colocalization in astrocytes in acute SON slices during HOC, as assessed using immunohistochemistry and confocal microscopy.

HOC changed the distribution of GFAP filaments surrounding VP neurons (n = 8) (Figure 4A). There was an increased proportion of VP neurons surrounded by GFAP filaments (GVP/TVP =57.7 ± 9.3 % versus 32.6 ± 3.4% before HOC; *p* < 0.05; Figure 4A(b)). In the late HOC stage, 20 min, GFAP filaments retracted so that the proportion of VP neurons surrounded by GFAP filaments decreased significantly (27.6 ± 5.2%, *p* < 0.05, when compared with that at 5 min HOC; Figure 4A(b)), returning to the proportion recorded at 0 min HOC. Of note, this initial increase and subsequent decrease in neuronal coverage by GFAP filaments corresponds to morphological plasticity sequence, i.e., extension followed by retraction of astrocytic processes and, in parallel, HOC-evoked early cell volume increase followed by a subsequent, late RVD, respectively [5]. Nonetheless, in the presence of phloretin, 5 min HOC dramatically increased the proportion of neurons surrounded by GFAP filaments (77.5 ± 13.7% versus 39. 4± 8.0% at control; n = 5, *p* < 0.05; Figure 4Bb), which remained elevated at 20 min HOC (83.6 ± 6.2%, *p* < 0.05). This novel finding indicates that VRAC mediates the retraction of GFAP filaments during the late HOC stage.

To further study the effects that blocking VRAC with phloretin might have on AQP4 expression and its interaction with GFAP, we measured the intensity of AQP4 staining and its colocalization with GFAP in acute SON slices. As shown in Figure 4A, 5 min HOC transiently increased the intensity of AQP4 staining from 30.54 ± 1.90 before HOC to 61.14 ± 2.91, along with a trend of reversal decrease/subside at 20 min HOC (45.96 ± 4.15 at 20 min HOC), and the value at 20 min HOC is not significantly different from that at the initial condition and 5 min HOC (Figure 4A(c)). Phloretin (30 μmol/L) prevented the early HOC-evoked effect on AQP4 staining intensity (Figure 4B(c)). Moreover, 5 min HOC showed a trend of increasing the colocalization (MCC) of GFAP with AQP4 (67.0 ± 7.7% versus 44.5 ± 6.4% at 0 min HOC; Figure 4A(d)), with a significant reversal decrease at 20 min HOC (41.4 ± 3.9%, *p* < 0.05). In the presence of phloretin, HOC failed to change the extent of GFAP and AQP4 colocalization (Figure 4B(d)). These new findings suggest that VRAC plays a role in increased AQP4 expression in early HOC and in AQP4 interactions with GFAP in late HOC stages.

### 3.5. Effects of TGN-020 on LRRC8A Expression and Its Colocalization with GFAP in Acute SON Slices during HOC

Next, we tested whether AQP4 activity affects HOC-evoked changes in LRRC8A expression and its colocalization with GFAP in astrocytes in acute SON slices. As shown in Figure 5A, HOC significantly increased LRRC8A levels at 5 min and 20 min (101.80 ± 8.88 at 5 min HOC and 84.09 ± 8.70 at 20 min HOC versus 38.50 ± 3.62 at 0 min HOC, *p* < 0.001) (Figure 5A(b)). In the presence of 10 μmol/L TGN-020, an increase in LRRC8A staining intensity did not appear at 5 min HOC (35.03 ± 2.54 versus 49.13 ± 7.26 at 0 min HOC, *p* > 0.05), and it only occurred at the late HOC stage, 20 min, (112.20 ± 5.50 compared to 0 min HOC and 5 min HOC, *p* < 0.01, and *p* < 0.005, respectively) (Figure 5B(b)). The colocalization of GFAP with LRRC8A increased significantly to 56.8 ± 1.8% at 5 min HOC from 43.2 ± 4.2% before HOC (*p* < 0.05), followed by a trend of reversal decrease at 20 min HOC (47.4 ± 2.3%; *p* = 0.14), and the value at 20 min is not significantly different from that at the initial condition and 5 min HOC (Figure 5A(c)). In the presence of TGN-020, however, there was no significant change in the colocalization of GFAP with LRRC8A during the entire HOC period (Figure 5B(c)). This novel finding indicates that AQP4 activity plays a role in early HOC-evoked changes in astrocytes, i.e., both increases in LRRC8A/VRAC expression and in its colocalization with GFAP.

### 3.6. Effects of HOC on VRAC Currents in Cultured Hypothalamic Astrocytes

The results presented above (Figure 1 and Figure 2) support the involvement of astrocytic VRAC and AQP4 in the HOC-evoked rebound excitation of VP neurons at late HOC stages. However, the dynamics of the VRAC activity in astrocytes and possible modulation by AQP4 channels during HOC are unknown. To address this issue, we recorded VRAC currents from individual cultured hypothalamic astrocytes at 13 days in culture in the whole-cell patch-clamp configuration. First, we pharmacologically confirmed the identity of electrophysiologically isolated VRAC currents using phloretin. At rest, astrocytes displayed VRAC currents, which were augmented at 10 min HOC (8.32 ± 1.71 of I_HOC 0 min_) and subsequently dampened by the application of 30 μmol/L phloretin (4.95 ± 1.38 of I_HOC 0 min_, Figure 6A). When HOC was applied to astrocytes pre-treated with 30 μmol/L phloretin, the HOC failed to augment VRAC currents (n = 6; 3.55 ± 1.47 of I_CTR_ in phloretin + 10 min HOC; Figure 6B). Thus, our electrophysiology approach reports on HOC-activated outwardly rectifying currents that are bona fide VRAC currents. Of note, HOC times of 1.5 min and 10 min in cell culture correspond to 5 and 20 min in acute slices, respectively; these times represent early and late stages of HOC, respectively [11].

Next, we assessed whether VRAC currents in hypothalamic astrocytes might be modulated by the activity of AQP4 channels during HOC. Again, astrocytes at rest displayed VRAC currents, the amplitude of which was augmented by HOC at 10 min (9.51 ± 1.57 of I_HOC 0 min_). This effect was not affected by subsequent application of 10 μmol/L of TGN-020 (11.05 ± 2.44 of I_HOC 0 min_; n = 12, *p* < 0.05; Figure 7A), indicating a lack of role for AQP4 in the modulation of VRAC once they are already activated by HOC. Pre-application of TGN-020 before HOC had no significant effect on VRAC currents in astrocytes (2.72 ± 0.49 of I_CTR_); however, it did not allow the development of HOC-evoked VRAC currents at 10 min (3.781 ± 0.972 of I_CTR_; n = 6; Figure 7B), indicating that AQP4 activity is necessary for the HOC-evoked activation of VRAC.

### 3.7. TGN-020 or Phloretin Distinctly Affects HOC-Evoked Changes in Astrocytic Volume in Cell Culture

In SON slices, we have previously identified that HOC dually regulates astrocytic morphological plasticity based on GFAP expression and spatial distribution [7]; GFAP filament distribution mirrors astrocyte volume transfer between soma and distal processes [5]. However, it is unknown whether hypothalamic astrocyte volemic changes and their related HOC-evoked dual morphological plasticity are associated with the concerted activities of VRAC and AQP4.

To address this issue, we first assessed the effects of TGN-020 or phloretin on astrocytic volemic changes. We loaded cultured hypothalamic astrocytes with SR101 and measured the changes in somata fluorescence intensity throughout HOC; swelling of astrocytes is seen as a decrease in fluorescence intensity. An inversion of relative change in fluorescence intensity represents changes in astrocytic somatic volume (∆V/V_0_). As shown in Figure 8, HOC caused swelling of astrocytic somata with a 45.3 ± 2.3% volemic increase in the first 1.5 min (n = 43, *p* < 0.005), which was followed by dramatic RVD (astrocytic volume decrease to 11.8 ± 3.6%, *p* < 0.005) at 10 min HOC (Figure 8A).

Next, we tested the effects of TGN-020 (10 µmol/L) on HOC-evoked astrocytic volemic changes. In the presence of TGN-020, astrocytes showed delayed swelling (Figure 8B), which developed late, at 10 min HOC, (25.9 ± 5.9%, *p* < 0.01 compared to both 0 and 1.5 min HOC), but not early, at 1.5 min (1.6 ± 2.2%, n = 14). Then, we tested the effects of blocking VRAC on HOC-evoked astrocytic volemic changes (Figure 8C). In the presence of 30 μmol/L of phloretin, astrocytes showed significant swelling at both 1.5 min HOC (47.1 ± 1.8%; n = 54, *p* < 0.005) and at 10 min HOC (27.6 ± 2.8%, *p* < 0.005) when compared to 0 min HOC.

Further comparison among groups showed that blocking AQP4 significantly reduced astrocytic swelling level at 1.5 min HOC (*p* < 0.01), while blocking AQP4 or VRAC both significantly hampered RVD (i.e., there was a higher relative volume change recorded) at 10 min HOC relative to that of the control (Figure 8D). These new findings indicate that both AQP4 and VRAC play vital roles in the occurrence of RVD in astrocytes, while AQP4 is critical for astrocyte swelling at the initial stage of HOC.

### 3.8. Effects of HOC, and Associated AQP4 and VRAC Activity, on Astrocytic GFAP and LRRC8A Distribution

The findings presented above (Figure 6, Figure 7 and Figure 8) indicate that both AQP4 and VRAC play an essential role in astrocytic volume regulation. Whether this process includes VRAC interactions with GFAP is unknown. To address this issue, we analyzed the spatial association of GFAP filaments with LRRC8A in the immunostaining of cultured hypothalamic astrocytes exposed to HOC. As shown in Figure 9A, GFAP filaments extended dramatically from 44.49 ± 1.58 μm at 0 min HOC to 75.62 ± 6.95 μm at 1.5 min HOC (*p* < 0.01), followed by a significant retraction to 31.27 ± 1.40 μm at 10 min HOC (*p* < 0.005 compared to 0 min and 1.5 min HOC; Figure 9A(a),A(b)). Similarly, LRCC8A levels increased significantly from 94.31 ± 11.17 at 0 min HOC to 168.40 ± 7.66 at 1.5 min HOC (*p* < 0.005), and then fell back (i.e., reversely decreased) to control levels at 10 min HOC (76.55 ± 6.40, *p* < 0.005 compared to 1.5 min HOC; Figure 9A(c)). Compared with 0 min HOC (76.9 ± 7.0%), the colocalization of these two molecules showed a trend of increasing at 1.5 min HOC (92.6 ± 2.4%) followed by a significant decrease at 10 min HOC (67.3 ± 7.2%, *p* < 0.05 compared to 1.5 min HOC; Figure 9A(d)). In the presence of 10 μmol/L of TGN-020, there was a delay in the HOC-induced increase in the length of the GFAP filaments. That is, there was no significant change at the early HOC stage of 1.5 min (48.32 ± 3.95 μm compared with 47.90 ± 3.49 μm at 0 min HOC), but an increase was evident at the late HOC stage of 10 min (69.46 ± 4.55 μm, *p* < 0.01) (Figure 9Bb). In the presence of 30 μmol/L phloretin, the length of the GFAP filaments progressively increased from 48.39 ± 2.97 μm before HOC to 77.34 ± 3.06 μm at 1.5 min HOC (*p* < 0.005) and then farther to 85.40 ± 5.93 μm at 10 min HOC (*p* < 0.005) (Figure 9C(b)). TGN-020 inhibited the reversal decrease in the LRRC8A level at 10 min HOC (132.00 ± 10.45 compared with 84.46 ± 19.29 at 0 min HOC) (Figure 9B(c)), while phloretin prevented an HOC-induced increase in LRRC8A levels (71.86 ± 6.02, 62.68 ± 4.03, and 63.37 ± 6.28 at 0, 1.5, and 10 min HOC, respectively; Figure 9C(c)). Moreover, in the presence of TGN-020 (Figure 9B(d)) or phloretin (Figure 9C(d)), GFAP-LRRC8A colocalization dynamics were suppressed, as a decrease in the colocalization seen in the control at 10 min HOC vanished. Taken together, these results, based on pharmacological intervention, indicate that GFAP and LRRC8A/VRAC dynamics rely on the activity of AQP4 and/or VRAC. Specifically, AQP4 activity is essential for the HOC-evoked early extension and late retraction of GFAP filaments, as well as for the HOC-evoked late reversal of LRRC8A/VRAC levels. VRAC activity is important for (i) the HOC-evoked late retraction of GFAP filaments during the RVD (but not for the early extension of GFAP filaments) and (ii) for its own (LRRC8A) HOC-evoked initial increase. Both VRAC and AQP4 activities are needed in the late decrease in colocalization between GFAP and LRRC8A.

## 4. Discussion

The present study reveals that the dual HOC effect on VP neuronal firing activity is not only associated with the expression and activities of AQP4, but also with those of LRRC8A. It reveals that these two molecules are molecularly associated and mutually dependent in HOC-evoked astrocytic RVD. Lastly, the normal activity of LRRC8A or AQP4 differently modulates GFAP filament length and astrocytic volume. This work provides not only a new and comprehensive view of the interactions between VRAC/LRRC8A, AQP4, and GFAP during HOC, but also generates additional evidence supporting the guiding/nodal role of GFAP [5] in astrocyte morpho-functional plasticity.

### 4.1. Dual HOC Effect on Astrocytic Plasticity and VP Neuronal Activity

It has been well established that in the SON of the hypothalamus, HOC can decrease VP neuronal activity and VP secretion early on (corresponding here to 5 min in acute slices); however, this initial effect reverses during prolonged HOC (corresponding here to 20 min in acute slices), resulting in a rebound excitation [6]. The present findings (Figure 1 and Figure 2) confirm these reports and further highlight the importance of AQP4 (Figure 1) and VRAC (Figure 2) in the HOC-evoked modulation of VP neuronal activity. This dual effect is associated with a change in intracellular osmotic pressure [30] and astrocytic morphological and functional plasticity, the latter mainly mediated by GFAP as a nodal molecule.

During HOC, the reduction in extracellular osmolality creates an osmotic pressure gradient across the plasma membrane and provides a driving force for water influx [5]. It is the abundant expression of AQP4 on the astrocytic plasma membrane that allows water influx during HOC, which causes an expansion of astrocytic volume (Figure 8) and decreases cytosolic ionic concentration [14]. Increased astrocytic volume, particularly the expansion of astrocytic processes around VP neurons (Figure 4 and Figure 5), strengthens physical barriers between adjacent VP neurons while accelerating the uptake of extracellular K^+^ and glutamate released during neuronal activity, thereby decreasing VP neuronal activity [5].

During late HOC, as the cytosolic ionic concentration decreases, RVD initiates [31]. RVD can result in the release of Cl^−^, glutamate, and aspartate from astrocytes through VRAC alongside water efflux [32] and a reduction in astrocytic volume, which can form the basis of the rebound excitation of VP neurons and VP hypersecretion in acute hyponatremia, as previously reviewed [5]. In the present study, this notion is further supported by the blocking effect of TGN-020 (Figure 1B) or phloretin (Figure 2) on the HOC-evoked rebound increase in VP neuronal activity and on the RVD of cultured astrocytes (Figure 8).

### 4.2. HOC Effect on sEPSC Frequency and Role of Astrocytes

HOC can modulate VP neuronal activity through the reduction in sEPSC frequency, in which AQP4-associated astrocytic swelling participates in the modulation in the early HOC stage while VRAC is involved in both the early and late stages of HOC reduction in sEPSC frequency (Figure 3). It is likely that VRAC’s contribution to the rebound occurs via the extrasynaptic NMDA receptor [6]. Namely, the occurrence of RVD increases glutamate spillover because of the withdrawal of astrocytic perisynaptic processes that highly express plasmalemmal glutamate transporters. This leads to a reduction in glutamate uptake from the synaptic cleft, and this transmitter can diffuse/spill into the extrasynaptic space, which can lead to increased activation of extrasynaptic NMDA receptors [33].

In astrocyte-specific LRRC8A knockout mice, glutamatergic transmission in the hippocampus is impaired due to a decrease in presynaptic release probability and in ambient glutamate level [34]. This report seems to be at odds with our finding that the inhibition of VRAC hampers the HOC-mediated inhibition of sEPSC frequency. However, this discrepancy is likely due to the time domain, as the HOC effect on sEPSCs via VRAC/LRRC8A in acute hippocampal slices lasted only several minutes before it turned into long inhibition [34]. Our observation of HOC’s effect in acute SON slices lasted 20 min, allowing time for RVD to occur. Thus, it is likely that blocking VRAC reduces astrocytic glutamate release through these anionic channels [35] and thus increases glutamate availability for synaptic glutamate release, via the astrocyte-neuron glutamine-glutamate cycle, but reduces extrasynaptic NMDA receptor activation. However, this speculation needs further investigation. Furthermore, we recorded a trend of reduction in sEPSC amplitude in response to HOC, which appeared to be blocked by blocking VRAC. Further study recording miniature EPSCs should provide more insight into pre- versus post-synaptic modulation by astrocytic VRAC.

### 4.3. Dual HOC Effect on GFAP Plasticity and Associated APQ4 and/or VRAC Activity

During neuronal activity in the SON, GFAP exhibits morphological plasticity endowed by its quick changes in the (dis)assembly, catabolism and (de)polymerizion [5]. Moreover, GFAP may serve as a node/guide for the transportation and installation of plasma membrane proteins in astrocytes, thereby determining astrocytic functional plasticity [5]. These characteristics in the SON have been summarized in a recent review [5] and supported by recent reports [7,11]. In the present study, we found changes in the colocalization of GFAP with LRRC8A in astrocytes exposed to HOC (Figure 5 and Figure 9). In its guiding/nodal role, it is likely that by pulling more LRRC8A together during HOC, GFAP can coordinate the activity of VRAC. Consequently, the extension of GFAP filaments promotes water influx and organic anion efflux in the early stage of HOC, while the retraction of GFAP filaments pulls the plasma membrane on astrocytic distal, perisynaptic processes toward soma and proximal processes and thus drives outflows of Cl^−^, organic anions (e.g., glutamate), and water, which initiates RVD. It should be noted that HOC in SON acute slice GFAP-LRRC8A colocalization shows an early increase with only a trend in late reversal decrease (Figure 5A(c)), while the opposite occurs in cultured hypothalamic astrocytes exposed to HOC (Figure 9A(d)). Another difference between the experimental models is evident in the persistent and late HOC-evoked increase in LRRC8A levels in acute slices (Figure 5A(b)), while in cultures astrocytes, there is a late reduction in LRRC8A levels (Figure 9A(c)). These differences should not distract from the fact that both AQP4 and VRAC play a role in LRRC8A expression levels and its colocalization with GFAP.

### 4.4. Role of AQP4 in Dual HOC Effect on Astrocytic Volume

In astrocytic morphological plasticity in the SON, the activation of AQP4 is essential for both the initial astrocyte swelling and for the later RVD in HOC, as shown in Figure 9. At different stages of HOC, AQP4 plays different roles. During the extension of GFAP filaments (Figure 4 and Figure 5) and expansion of astrocytic volume (Figure 8), the association between GFAP filaments and AQP4 begins to increase (seen as a trend in Figure 4A(d)), while the plasma membrane localization and functions of AQP4 also increase significantly [5,7]. These events secure water influx promptly to support the volume increase, since TGN-020 blocks astrocyte swelling (Figure 8B). Once cytosolic ionic strength is reduced with additional water entry, the reversal osmotic pressure across the plasma membrane begins to remove excessive water through AQP4 during RVD [10]. This inference is supported by the blocking effect of TGN-020 pretreatment on RVD (Figure 9D) and the HOC-evoked augmentation of VRAC currents at 10 min HOC (Figure 7).

### 4.5. Role of VRAC in Dual HOC Effect on Astrocytic Volume and VP Neuronal Activity

In the present study, LRRC8A was identified in astrocytes in acute SON slices (Figure 5) and in cultured hypothalamic astrocytes (Figure 9). The HOC-evoked rebound of VP neuronal activity at the late stage of HOC was blocked by either phloretin or DCPIB (Figure 2), blockers of VRAC currents. We subsequently used only phloretin, which blocked the HOC-evoked augmentation of VRAC currents at the late stage of HOC (Figure 6). Furthermore, phloretin blocked HOC-evoked RVD in cultured hypothalamic astrocytes (Figure 8). Thus, HOC-evoked intracellular hypotonicity and swelling- triggered VRAC currents are involved in the generation of RVD.

In the SON, basal and evoked taurine release during RVD are strongly inhibited by the glia-specific toxin fluoroacetate [36], indicating that the VRAC-associated neurochemical events should be of astrocytic origin. Thus, the HOC-evoked rebound excitation of VP neurons should result from the activation of astrocytic VRAC. It should be noted that in the present work, we used cultured hypothalamic astrocytes (a subset of which originated from the SON) that may collectively display some features different from SON astrocytes. Our future work should take an arduous approach to study cultured astrocytes from the SON alone.

### 4.6. Interactions of Volume-Regulating Channles in RVD: Translational Relevance

Our findings on the role of VRAC and AQP4 in the astrocytic modulation of VP neuronal activity in the SON, as a consequence of astrocytic RVD and interactions of VRAC with GFAP, could have translational value in the treatment of, e.g., ischemic brain edema and hydromineral imbalance [37]. As a multi-targeted therapeutic approach may prove necessary to treat these neurological disorders, other volume-regulating channels [38], such as TRPV4 channels [12] and Kir4.1 [11], need to be taken in account. Their likely interplay with AQP4 and/or VRAC in HOC should be studied, along with defining the relative contributions of each of the volume-regulating channels in RVD and HOC.

Intramembrane organization and/or protein isoforms, e.g., M1 and M23 AQP4 isoforms, may have influenced our results. As changes in the M1 to M23 ratio may affect AQP4 function [39], the question arises as to whether this isoform ratio matters in the outcome of RVD and the modulation of neuronal activity. Could there be agents/drugs that can modulate this ratio and, in turn, might they be therapeutically useful? Clearly, these questions are out of scope of the present study, but they represent our future endeavors.

An additional area of future interest is studying the interactions of additional volume-regulating channels with GAFP, given the guiding role of GFAP in astrocyte plasticity [5]. It may well be that the interruption of interactions between individual types of, or a combination of, volume-regulating channels and GFAP emerges as a fertile ground for therapeutic intervention. Furthermore, as GFAP is involved in organelle trafficking in astrocytes [5,40], it may be that this intermediary filament governs the trafficking/cycling of volume-regulated channel-laden organelles (i.e., the cytosolic pool) to/from the plasmalemma. If so, the modulation of cycling parameters may represent an additional site for therapeutic intervention, as the effect of preventing/enhancing the delivery of VRAC to the plasma membrane or its retrieval from the plasma membrane would affect HOC, RVD, and the modulation of VP neuronal activity in the SON.

## 5. Conclusions

The interplay between GFAP, AQP4, and VRAC is important for astrocytic volume regulation, in which GFAP likely plays a guiding role for AQP4 and VRAC, while the activity of the latter two molecules provides a permissive role in GFAP plasticity. In astrocytic volume regulation during HOC, AQP4 is involved both in cell swelling and in RVD under the drive of VRAC-dependent transmembrane osmotic gradients, while VRAC mainly works in RVD following the activation of these channels via AQP4-dependent intracellular hypotonicity. Further study on the interactions between GFAP, AQP4, and VRAC and other volume-regulating molecules in astrocytic volemic regulation is warranted in order to establish the entire functional molecular network responsible for astrocytic plasticity and to understand the astrocytic regulation of neuronal activity.

## Figures and Tables

**Figure 1 cells-12-01723-f001:**
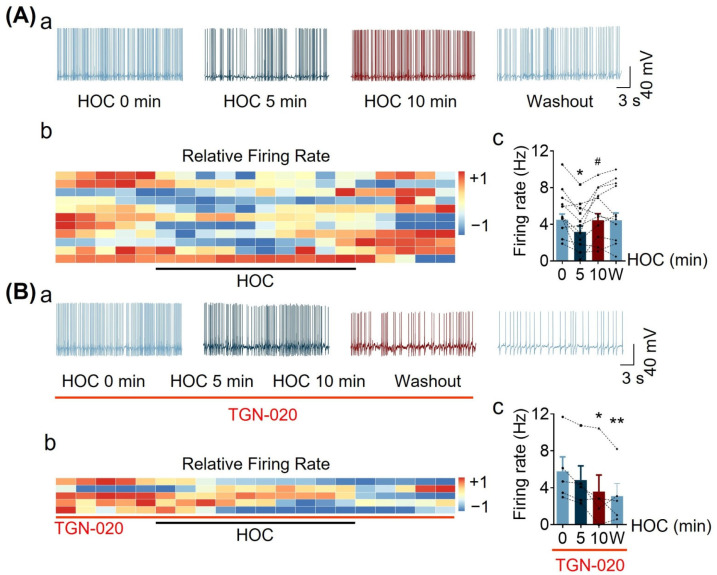
Effect of TGN-020, an aquaporin 4 (AQP4) blocker, on hyposmotic challenge (HOC) -modulated vasopressin (VP) neuronal firing activity in acute supraoptic nucleus (SON) slices. (**A**) Time course of changes in the firing rate of VP neurons using whole-cell patch-clamp recordings. (**a**) Representative episodes of VP neuronal firing activity at different stages of HOC. (**b**) Heatmap of the relative firing rate of VP neurons at different stages of HOC (+1 represents the highest frequency of firing rate, −1 represents the lowest frequency of firing rate; each strip represents a recording from a single neuron). (**c**) Bar graphs showing average VP neuronal firing rates at different stages of HOC. Data from individual neurons are shown by line-connected points. Abscissa indicates time (in minutes of HOC) (n = 11; *, *p* < 0.05 compared with 0 min HOC; #, *p* < 0.05 compared with 5 min HOC; ANOVA, Sidak). (**B**) Effect of TGN-020 (10 μmol/L) on the time-dependent changes in the firing rate of VP neurons under HOC in representative recording episodes. (**a**) Heatmap of relative firing rate (**b**) and summary bar graphs ((**c**), n = 5; *, *p* < 0.05, **, *p* < 0.01 compared with 0 min HOC; ANOVA, Sidak). Abbreviations: W, washout (isosmotic condition).

**Figure 2 cells-12-01723-f002:**
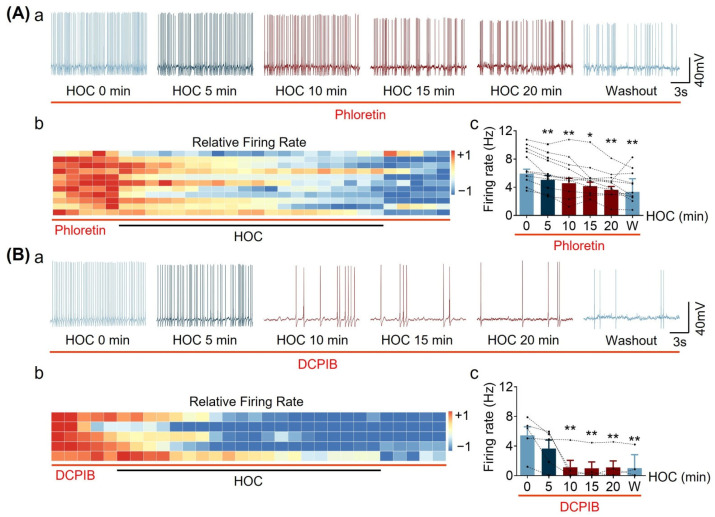
Effect of volume-regulated anion channel (VRAC) blockers, phloretin or DCPIB, on HOC-modulated electrical activity of VP neurons in acute SON slices. (**A**). Changes in the HOC-mediated time-dependent firing rate of VP neurons in the presence of 30 μmol/L phloretin in whole−cell patch-clamp recordings: (**a**) representative episodes of VP neuronal firing activity; (**b**) heatmap of the relative firing rate; and (**c**) bar graphs of average VP neuronal firing rates and data from individual neurons shown by line-connected points (n = 11; *, *p* < 0.05; **, *p* < 0.01 compared with 0 min HOC; ANOVA, Bonferroni. (**B**). Changes in the HOC-modulated time-dependent firing activity of VP neurons in the presence of 20 μmol/L DCPIB: (**a**–**c**), annotation as in (**A**); ((**c**), n = 5; **, *p* < 0.01, compared with 0 min HOC; ANOVA, Sidak). For other annotations and abbreviations, refer to Figure 1.

**Figure 3 cells-12-01723-f003:**
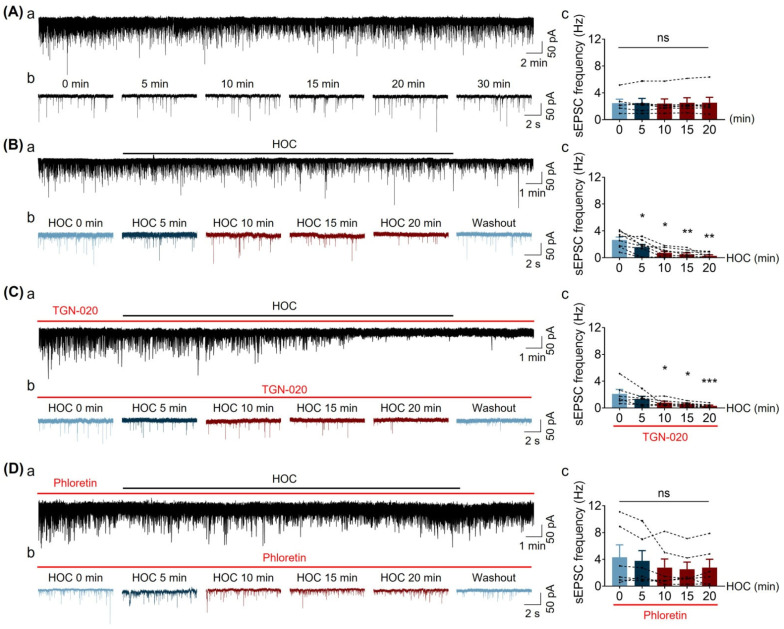
Effect of TGN-020 or phloretin on spontaneous excitatory postsynaptic current (sEPSC) frequency of VP neurons during HOC in acute SON slices. (**A**). Representative sEPSCs from VP neurons at rest recorded in the whole-cell voltage clamp mode in continuous recording ((**a**), Vh= −70 mV) or in recording episodes (**b**) and summary of average sEPSC frequency ((**c**), n = 6; ANOVA, Bonferroni). (**B**). sEPSCs from VP neurons at various time points of HOC in continuous recording or in recording episodes (**a**,**b**), respectively and summary of average sEPSC frequency ((**c**), n = 8; *, *p* < 0.05; **, *p* < 0.01 compared with 0 min HOC; ANOVA, Bonferroni). (**C**). sEPSCs from VP neurons at various time points during HOC (**a**,**b**) and summary of average sEPSC frequency ((**c**), n = 6; *, *p* < 0.05; ***, *p* < 0.005 compared with 0 min; ANOVA, Bonferroni) in the presence of 10 μmol/L TGN-020. (**D**). sEPSCs from VP neurons at various time points during HOC (**a**,**b**) and summary of average sEPSC frequency ((**c**), n = 6; ANOVA, Bonferroni) in the presence of 30 μmol/L phloretin. Total duration of each recording from individual cells in (**A**)a, (**B**)a, (**C**)a, and (**D**)a is 40 min. Abbreviations: sEPSC, excitatory postsynaptic currents; ns, non-significant. For other annotations and abbreviations, refer to Figure 1.

**Figure 4 cells-12-01723-f004:**
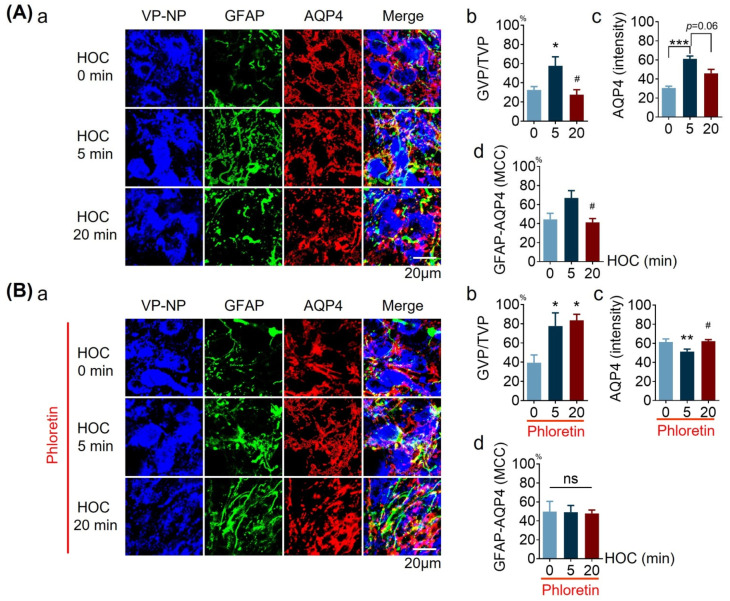
Effects of phloretin on GFAP filament distribution, AQP4 expression, and their colocalization during HOC in acute SON slices. (**A**,**B**). Confocal microscopy of immunohistochemistry: (**a**) VP-NP (blue), GFAP (green), AQP4 (red), and merged images at different stages of HOC without (**A**) and with 30 μmol/L phloretin (**B**). Bar graphs summarizing the proportion of VP neuronal somata surrounded by GFAP filaments (GVP) relative to total number of VP neurons (TVP, (**b**)), the staining intensity of AQP4 (**c**), and the MCC for GFAP and AQP4 (**d**), respectively. Scale bars =20 µm; *, *p* < 0.05 compared with 0 min; HOC #, *p* < 0.05; **, *p* < 0.01; ***, *p* < 0.005 compared with 5 min HOC; ANOVA (Bonferroni for (**A**)b, (**B**)b and (**B**)d; Dunn for (**A**)c, (**A**)d and (**B**)c). Abbreviations: AQP4, aquaporin 4; GFAP, glial fibrillary acidic protein; MCC, Manders’ colocalization coefficient; ns, non-significant; VP, vasopressin; VP-NP, VP neurophysin stain; other annotations as in Figure 1.

**Figure 5 cells-12-01723-f005:**
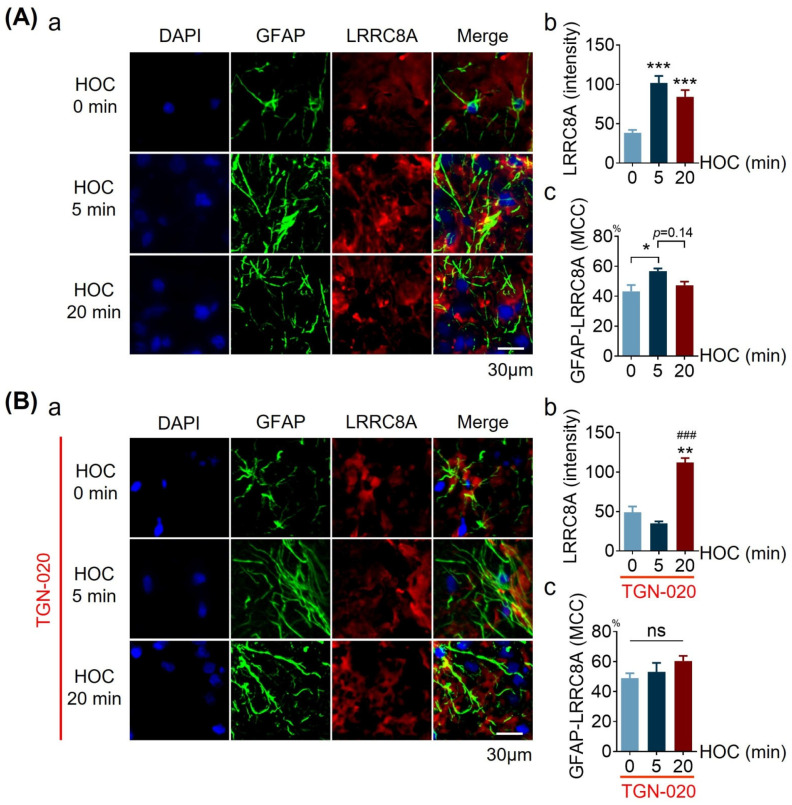
Effects of blocking AQP4 on HOC-evoked increase in LRRC8A expression and its colocalization with GFAP in acute SON slices. (**A**,**B**). Confocal microscopy of immunohistochemistry of nuclei (DAPI; blue), GFAP (green), and LRRC8A (red) in (**a**), and at different times of HOC without (**A**) and with 10 μmol/L TGN-020 (**B**). Bar graphs summarizing the staining intensity of LRRC8A (**b**) and the MCC for GFAP and LRRC8A (**c**) at different times of HOC. Scale bars = 30 µm; in A, n = 6; *, *p* < 0.05, **, *p* < 0.01, and ***, *p* < 0.005 compared with 0 min HOC (ANOVA, Dunn); in (**B**), n = 10; ***, *p* < 0.005 compared with 0 min HOC; ###, *p* < 0.001 compared with 5 min HOC (ANOVA, Dunn). Abbreviations: LRRC8A, leucine-rich repeat-containing protein 8A; other annotations as in Figure 1 and Figure 4.

**Figure 6 cells-12-01723-f006:**
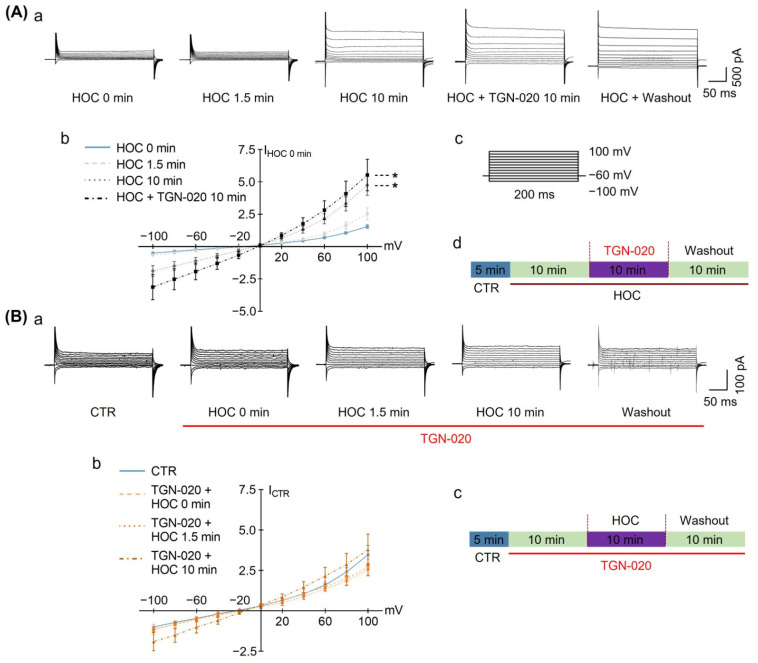
Effects of phloretin on HOC-evoked astrocytic VRAC currents. (**A**). HOC-evoked VRAC currents in astrocytes at late HOC stages (**a**) are dampened by subsequent application of 30 μmol/L of phloretin. (**b**) Relative current−voltage curves obtained from astrocytes under different conditions normalized to currents recorded just prior to the HOC onset (I_HOC 0 min_; n = 7; *, *p* < 0.05, compared with 0 min HOC; #, *p* < 0.05, compared with 10 min HOC; ANOVA, Bonferroni). (**c**) A series of voltage steps, initiated from a holding potential of −60 mV, were used to test for VRAC currents in astrocytes. (**d**) Schematics of liquid perfusion protocol. (**B**). Pre-application of 30 μmol/L phloretin occludes the development of HOC-evoked VRAC currents in astrocytes (**a**). (**b**) Relative current-voltage curves obtained from astrocytes under different conditions, normalized to current at control conditions (I_CTR_; n = 6; ANOVA, Bonferroni). (**c**) Schematics of liquid perfusion protocol. Other annotations as in Figure 1.

**Figure 7 cells-12-01723-f007:**
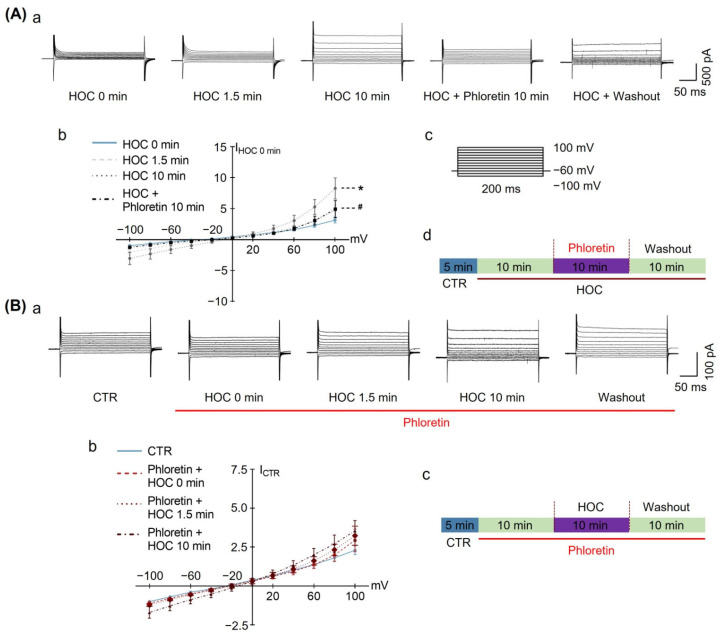
Effects of TGN-020 on HOC-evoked VRAC currents in cultured hypothalamic astrocytes. (**A**). (**a**) HOC-evoked VRAC currents in astrocytes, after developing HOC for 10 min, are not affected by subsequent application of 10 μmol/L TGN-020; (**b**) Current-voltage curves of outwardly rectifying VRAC currents obtained from astrocytes under different conditions, normalized to currents recorded just prior to the HOC onset (I _HOC 0 min_; n = 12; *, *p* < 0.05, compared with 0 min HOC; #, *p* < 0.05, compared with 10 min HOC; ANOVA, Bonferroni). (**c**) A series of voltage steps, initiated from a holding potential of −60 mV, were used to test for VRAC currents in astrocytes. (**d**) Schematics of liquid perfusion protocol. (**B**). (**a**) Pre-application of 10 μmol/L of TGN-020 occluded development of HOC-evoked VRAC currents in astrocytes. (**b**) Current-voltage curves obtained from astrocytes under different conditions, normalized to current in control conditions (I_CTR_) (n = 6; ANOVA, Bonferroni). (**c**) Schematics of liquid perfusion protocol. Other annotations as in Figure 1 and Figure 6.

**Figure 8 cells-12-01723-f008:**
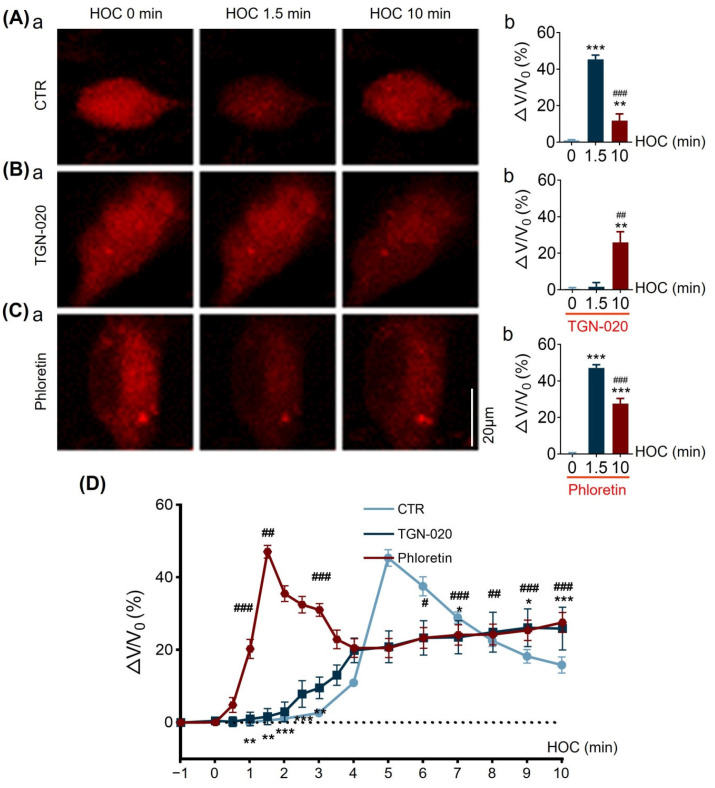
TGN-20 or phloretin causes a distinct effect of HOC-evoked volemic changes in cultured hypothalamic astrocytes. (**A**–**C**). Fluorescence images of sulforhodamine 101 (SR101)-loaded astrocytic somata showing fluorescence intensity decrease, which report on relative volume increase (∆V/V_0_) during HOC under control conditions (CTR; (**A**)), or in the presence of 10 μmol/L of TGN-020 (**B**) or 30 μmol/L of phloretin (**C**) at different time points (**a**). The corresponding summarizing bar graphs showing average changes (**b**). TGN-020 or phloretin is applied at 1 min prior to HOC and then kept throughout HOC. Scale bar = 20 μm; CTR, n = 43; TGN-020, n = 14; phloretin, n = 54; **, *p* < 0.01; ***, *p* < 0.005, compared with 0 min HOC; ##, *p* < 0.01; ###, *p* < 0.005, compared with 1.5 min HOC; ANOVA, Dunn. (**D**) Summary of time lapse of volume changes showing distinct effects of TGN-020 (*, *p* < 0.05; **, *p* < 0.01; ***, *p* < 0.005 compared with CTR; ANOVA, Sidak) or phloretin (#, *p* < 0.05; ##, *p* < 0.01; ###, *p* < 0.005 compared with CTR; ANOVA, Sidak). Dotted line in extrapolation of the initial volume in CTR.

**Figure 9 cells-12-01723-f009:**
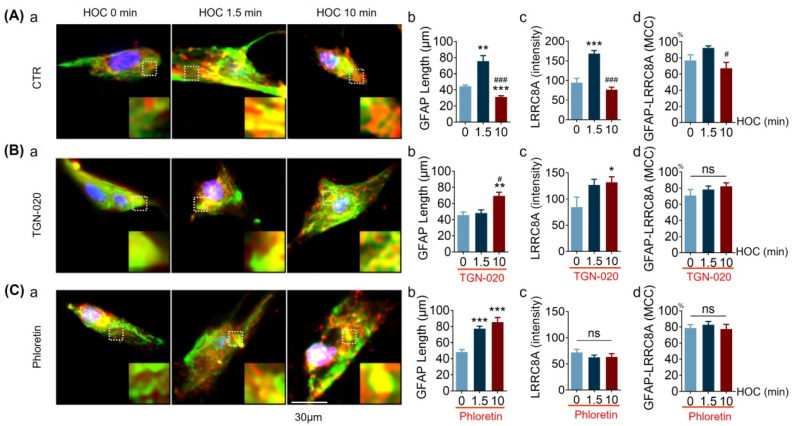
TGN-20 or phloretin cause distinct effects on HOC-evoked GFAP filament length, LRRC8A expression, and GFAP-LRRC8A colocalization in cultured hypothalamic astrocytes. (**A**–**C**). Representative fluorescence images of GFAP (green), LRRC8A (red), and nuclei (blue, DAPI) at 0, 1.5, and 10 min HOC (**a**) in CTR ((**A**); n = 20, 15, and 10, respectively), 10 μmol/L of TGN-020 ((**B**); n = 17, 26, and 23, respectively), or 30 μmol/L of phloretin ((**C**); n = 20, 39, and 29, respectively). GFAP filament length (**b**), expression, i.e., staining/fluorescence intensity, of LRRC8A (**c**), and the MCC of GFAP with LRRC8A (**d**), respectively. Scale bar = 30 µm; *, *p* < 0.05, **; *p* < 0.01, ***, *p* < 0.005 compared with 0 min HOC; #, *p* < 0.05, ###, *p* < 0.005 compared with 1.5 min HOC; ANOVA, (Bonferroni for (**C**)b; Sidak for (**A**)c, (**B**)c, and (**C**)c; Dunn’s for (**A**)b, (**A**)d, (**B**)b, (**B)d**, and (**D**)d). Other annotations as in Figure 5.

**Table 1 cells-12-01723-t001:** Summary of early and late HOC effects and associated AQP4 and/or VRAC supporting roles.

			Effect	Supporting Role		Effect	Supporting Role	
	Measurements	Figure	Early HOC	AQP4	VRAC	Late HOC	AQP4	VRAC
Acute Slice	Firing rate	Figure 1 and Figure 2	↓	Y	Y	R	Y	Y
	sEPSC frequency	Figure 3	↓	Y	Y	↓	N	Y
	GVP/TVP	Figure 4	↑	NT	N	R	NT	Y
	AQP4 intensity/level	Figure 4	↑	NT	Y	R, t	NT	ND-t
	GFAP-AQP4 MCC	Figure 4	↑, t	NT	ND-t	R	NT	Y
	LRRC8A intensity/level	Figure 5	↑	Y	NT	↑	N	NT
	GFAP-LRRC8A MCC	Figure 5	↑	Y	NT	R, t	ND-t	NT
Astro Culture	VRAC current	Figure 6 and Figure 7	NC	NA	NA	↑	N; Y-pt	Y
	∆V/V_0_	Figure 8	↑	Y	N	RVD	Y	Y
	GFAP filament length	Figure 9	↑	Y	N	R	Y	Y
	LRRC8A intensity/level	Figure 9	↑	Y	Y	R	Y	N
	GFAP-LRRC8A MCC	Figure 9	↑, t	ND-t	ND-t	↓	Y	Y

NB: early HOC refers to 5 min in acute slices and 1.5 min in cultured astrocytes; late HOC refers to 10 min in acute slices and 5 min in cultured astrocytes. Abbreviations of table entries/symbols: N, no; NA, not applicable; NC, no change; ND-t, not determined due to a trend in an HOC effect; NT, not tested; pt, pretreatment; R, rebound, retraction, or reversal decrease/subside; RVD, regulatory volume decrease; t, trend; Y, yes; ↓, decrease; ↑, increase. Abbreviations of terms: AQP4, aquaporin 4; Astro, astrocyte; GFAP, glial fibrillary acidic protein; GVP, VP neuronal somata surrounded by GFAP filaments; HOC, hyposmotic challenge; LRRC8A, leucine-rich repeat-containing protein 8 A; MCC, Manders’ colocalization coefficient; sEPSC, spontaneous excitatory postsynaptic current; VRAC, volume-regulated anion channel; TVP, total number of VP neurons; △V/V_0_, relative volume change.

## Data Availability

All data generated or analyzed during this study are included in this article. Further inquiries can be directed to the senior corresponding author.

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
