# Peer review of "Interactions between the Astrocytic Volume-Regulated Anion Channel and Aquaporin 4 in Hyposmotic Regulation of Vasopressin Neuronal Activity in the Supraoptic Nucleus"

_cells, 2023, doi:10.3390/cells12131723_

Round 1

Reviewer 1 Report

In their study Liu and colleagues show how volume-regulated anion channels and aquaporin 4 interact following hyposmotic stimuli, and more interestingly, that through this interaction, the neuronal activity is significantly regulated. The study is robust and convincing in its rationale, methods, and conclusions. The quality of the results and the figures are high. The experimental data fully support the main findings on the effects on neuronal excitability. 

As a minor comment, the data should also be discussed in light of Glu dynamics and its spillover considering the astrocytic processes highly expressed Glu transporters.

Author Response

Review 1: In their study Liu and colleagues show how volume-regulated anion channels and aquaporin 4 interact following hyposmotic stimuli, and more interestingly, that through this interaction, the neuronal activity is significantly regulated. The study is robust and convincing in its rationale, methods, and conclusions. The quality of the results and the figures are high. The experimental data fully support the main findings on the effects on neuronal excitability. As a minor comment, the data should also be discussed in light of Glu dynamics and its spillover considering the astrocytic processes highly expressed Glu transporters.

Response: We thank reviewer for his insightful comments and kind words. To address the issue of glutamate dynamics as pertains to our findings, we have added text in the discussion (ln 602-606, page 21, highlighted in yellow).

Reviewer 2 Report

In the work by Yang Liu and colleagues, the authors aimed to analyze the interaction of volume-regulated anion channels (VRACs) with an astrocyte-specific water channel aquaporin 4 (AQP4) in response to hypo-osmotic challenge (HOC) using rat brain slices containing the supraoptic nucleus (SON) and primary cultures of hypothalamic astrocytes.
While the study’s aim is interesting and actual, the manuscript content doesn’t help to find strict novelty. Indeed, experiments are well-performed with meticulously collected data. The reader is often referred to previously published data throughout the text, particularly in the description of the result section. The authors are advised to more clearly present what is new in the present study since most of the data, to some extent, recapitulates previous findings.
Specific comments:
1.    Since phloretin inhibitor presents indirect effects, it is unclear why, in the remaining experiments, a more selective VRAC inhibitor, DCPIB, with a more significant impact, wasn't used. Please provide a justification.
2.    The authors analyzed AQP4 immunofluorescence in studied experimental settings. However, the authors do not differentiate between cytosolic and membrane localization/distribution. How AQP4 intramembrane organization and/or protein isoforms may impact results interpretation?
3.    It is unclear why WB for AQP4/LRRC8A was performed. What are these protein half-lives?
4.    The importance, and translational relevance of the main finding(s), should be discussed in the context of other proteins involved in RVD (e.g., TRPV4, etc.) and their potential functional contribution/ and/or abundance/selectivity, etc.

Author Response

In the work by Yang Liu and colleagues, the authors aimed to analyze the interaction of volume-regulated anion channels (VRACs) with an astrocyte-specific water channel aquaporin 4 (AQP4) in response to hypo-osmotic challenge (HOC) using rat brain slices containing the supraoptic nucleus (SON) and primary cultures of hypothalamic astrocytes. While the study’s aim is interesting and actual, the manuscript content doesn’t help to find strict novelty. Indeed, experiments are well-performed with meticulously collected data. The reader is often referred to previously published data throughout the text, particularly in the description of the result section. The authors are advised to more clearly present what is new in the present study since most of the data, to some extent, recapitulates previous findings.

Response: We thank the reviewer for valuable comments. We have now revised the manuscript to address the concerns raised. We pruned the unnecessary density and overly extensive cross-referencing to previous work in the results sections. We also edited text for clarity. There  deletions and edits were left unmarked. We have now made additions, highlighted in turquoise, throrough the text to clarify the novelty of our present work (ln 73-76;235-236; 274-276;316-319; 328-330; 345-351; 382-383; 409-411; 423-425; 477; 480-482; 501; 520-522; 547-552; and 568-571, highlighted in turquoise). We also added Table 1 to summarize our findings.

Specific comments:

Response: Our response to the specific comments 1, 2 and 4 are highlighted in turquoise in the text. In regard to Specific Comment 3, here, we provide half-lives and the rationale why we elected to do WB at the first place, but now have removed them from the paper.

  1. Since phloretin inhibitor presents indirect effects, it is unclear why, in the remaining experiments, a more selective VRAC inhibitor, DCPIB, with a more significant impact, wasn't used. Please provide a justification.

Response: Well, we inadvertently reduced the significance of our work. Namely, all VRAC inhibitors have various off-target effects, regardless of their initially reported specificity. As this has been the case for phloretin, it has become perhaps a larger issue for DCPIB. One can successfully argued that DCPIB is a no better, if not subordinate, VRAC inhibitor when compared to phloretin, if one considers the extent to which DCPIB affects off targets.  However, the common inhibitory effects of two different VRAC inhibitors, with different, and to all we know non-overlapping, off target effects on HOC-evoked rebound in firing rate indicate that  activity of VRAC is essential for the rebound of VP neurons firing rate. Subsequently to this data obtained using our most sensitive assay, we elected to use only one inhibitor. Regardless which one we have chosen, based on concentrations we used and known off-targets, either would be a justifiably choice. We gave an edge to phloretin mainly due to our perception of a lesser known number of off-targets affected. We now clearly outline in 3.2 section of result (ln 300-301 and 305-313) the rationale for using two blockers in electrophysiology and subsequent use of one/ phoretin.

  1. 2. The authors analyzed AQP4 immunofluorescence in studied experimental settings. However, the authors do not differentiate between cytosolic and membrane localization/distribution. How AQP4 intramembrane organization and/or protein isoforms may impact results interpretation?

Response: The reviewer brings yet an excellent point, which we plan to address in our future studies. Meanwhile, to discuss this issue, along with the concern raised in the reviewer point #4, we have introduced the new subheading 4.6 in the discussion (ln 674-699). 

  1. It is unclear why WB for AQP4/LRRC8A was performed. What are these protein half-lives?

Response: This is an excellent point. The short answer is because we were curious, but were wrong to present those data. The half-lives of AQP4 and LRRC8A are about 8-9 hours [1]; [2] and thus simply measuring their expression levels in a short period may not be general accepted as meaningful.  However, if past (and present) findings on GFAP dynamics in the SON be our guide, we have found its dynamics unexpectedly fast on time scale of 10-15 minute, regardless of its rather long half-time of 24-48 h [3]. However, we do acknowledge that this might not be the case with AQP4 and LRRC8. Consequently, we generated seemingly disparate findings as our present data using Western blots (old Fig 4Ac, 4Cc for AQP4 and Fig 7A for LRRC8) are at odds with those results by immunochemistry (old Fig 4Cc, Dc and Fig. 10A, respectively); we observed only trends in Western blots and significant changes in immunochemistry. These discrepancies between Western blots and immunochemistry for proteins of long half-lives are common in the SON, and it is turning out that GFAP is the sole exception.  Rather that adding more reconciling statements in already crowded results and discussion sections, we removed that data. Consequently, we provide updated Fig. 4 and 7 lacking Western blots and removed Figure 6. This action does not change any outcome of the study, but rather makes presentation more streamlined. We do thank reviewer again to open our eyes. 

  1. The importance, and translational relevance of the main finding(s), should be discussed in the context of other proteins involved in RVD (e.g., TRPV4, etc.) and their potential functional contribution/ and/or abundance/selectivity, etc.

Reponses: The reviewer brings another excellent point to which we completely agree. As outlined in our response to the reviewer’s point #2, we have introduced the new subheading 4.6 in the discussion (ln 674-699). We also refined our statement in introduction (ln 63-64) to include TRP-V as a channel, activity of which contributes to osmotic gradient across the membrane. We also added in conclusions that other volume-regulating molecule’s need to be studied (ln 709)

References

  1. De Bellis, M.; Pisani, F.; Mola, M.G.; Basco, D.; Catalano, F.; Nicchia, G.P.; Svelto, M.; Frigeri, A. A novel human aquaporin-4 splice variant exhibits a dominant-negative activity: a new mechanism to regulate water permeability. Mol Biol Cell 2014, 25, 470-480, doi:10.1091/mbc.E13-06-0331.
  2. Chen, Y.; Zuo, X.; Wei, Q.; Xu, J.; Liu, X.; Liu, S.; Wang, H.; Luo, Q.; Wang, Y.; Yang, Y., et al. Upregulation of LRRC8A by m(5)C modification-mediated mRNA stability suppresses apoptosis and facilitates tumorigenesis in cervical cancer. Int J Biol Sci 2023, 19, 691-704, doi:10.7150/ijbs.79205.
  3. Papa, L.; Brophy, G.M.; Welch, R.D.; Lewis, L.M.; Braga, C.F.; Tan, C.N.; Ameli, N.J.; Lopez, M.A.; Haeussler, C.A.; Mendez Giordano, D.I., et al. Time Course and Diagnostic Accuracy of Glial and Neuronal Blood Biomarkers GFAP and UCH-L1 in a Large Cohort of Trauma Patients With and Without Mild Traumatic Brain Injury. JAMA Neurol 2016, 73, 551-560, doi:10.1001/jamaneurol.2016.0039.

Round 2

Reviewer 2 Report

The Authors sufficiently responded to all issues.